palaeontology/evolution/computational biology

Mesozoic birds, tip dating, relaxed clock, MrBayes

**Author for correspondence:**
Chi Zhang
e-mail: zhangchi@ivpp.ac.cn

# Bayesian tip dating reveals heterogeneous morphological clocks in Mesozoic birds

## Chi Zhang[1,2] and Min Wang[1,2]

[1]Key Laboratory of Vertebrate Evolution and Human Origins, Institute of Vertebrate Paleontology and Paleoanthropology, Chinese Academy of Sciences, Beijing 100044, People's Republic of China
[2]Center for Excellence in Life and Paleoenvironment, Chinese Academy of Sciences, Beijing 100044, People's Republic of China

CZ, 0000-0001-6009-5273; MW, 0000-0001-8506-1213

Recently, comprehensive morphological datasets including nearly all the well-recognized Mesozoic birds became available, making it feasible for statistically rigorous methods to unveil finer evolutionary patterns during early avian evolution. Here, we exploited the advantage of Bayesian tip dating under relaxed morphological clocks to estimate both the divergence times and evolutionary rates while accounting for their uncertainties. We further subdivided the characters into six body regions (i.e. skull, axial skeleton, pectoral girdle and sternum, forelimb, pelvic girdle and hindlimb) to assess evolutionary rate heterogeneity both along the lineages and across partitions. We observed extremely high rates of morphological character changes during early avian evolution, and the clock rates are quite heterogeneous among the six regions. The branch subtending Pygostylia shows an extremely high rate in the axial skeleton, while the branches subtending Ornithothoraces and Enantiornithes show notably high rates in the pectoral girdle and sternum and moderately high rates in the forelimb. The extensive modifications in these body regions largely correspond to refinement of the flight capability. This study reveals the power and flexibility of Bayesian tip dating implemented in MrBayes to investigate evolutionary dynamics in deep time.

## 1. Introduction

Birds are one of the most speciose (over 10 000 recognized species) and ecologically diverse living vertebrates [1]. The origin and evolution of birds have long been a hot debate in evolutionary biology, although it has been generally accepted that birds underwent two large-scale radiations during their 160 million years evolution, one for the stem groups in the Cretaceous and the other for the crown groups in the Paleogene [2–4]. Over the

last few years, numerous well-preserved Mesozoic bird fossils have been described [4,5], and consensus has been approached regarding their systematic relationships [6–8]. This wealth of data has significantly bridged the large morphological gap between birds and their non-avian theropod predecessors [4,9] and demonstrated their evolutionary success related to novel traits. Expanded morphological characters that cover the major Mesozoic avian groups with chronological data became accessible recently [10], making it possible to trace the early avian evolution more quantitatively and to address important questions such as the divergence times of the major clades and the patterns of morphological character changes within and between lineages [11,12]. However, few quantitative and statistical studies for early avian evolution have been performed as yet.

In a previous study [10], evolutionary rate heterogeneity in Mesozoic birds was investigated under maximum parsimony using a large morphological dataset containing 262 characters and 58 taxa [13]. The approach was stepwise: first to infer the most parsimonious trees, then to inform the internal node ages using certain ad hoc measures and last to obtain the branch rates from dividing the number of parsimonious changes by the time durations along the corresponding branches. It did not account for the uncertainties of the tree topology, times and rates statistically and was unable to model the evolutionary process explicitly. The dataset was then recently extended to 280 characters and 68 taxa [14], with newly recognized species and significantly revised morphological character scorings. Moreover, we revised this dataset on the basis of the most recent study [15], mainly focusing on the cranial morphology of Ichthyornithiformes and Hesperornithiformes. The modified dataset contains nearly all the well-recognized Mesozoic birds and represents the most comprehensive morphological characters hitherto, thus providing more power to unveil finer evolutionary patterns and becoming applicable to more statistically rigorous methods.

We further subdivided the characters into six anatomical regions to assess evolutionary rate heterogeneity across these regions: skull (53 characters), axial skeleton (36 characters), pectoral girdle and sternum (48 characters), forelimb (65 characters), pelvic girdle (23 characters) and hindlimb (55 characters). These partitions reflect relatively distinct body regions that have undergone different patterns of modification in the early phase of evolution. For example, the forelimb modified considerably when Ornithothoraces diverged from more stemward taxa and most of the changes pertain to the refinement of flight. The changes related to flight capability were also reflected in the pectoral girdle. The axial skeleton changed greatly, especially in the abbreviations of the tail from their dinosaurian ancestors, which placed the centre of gravity more anteriorly to be close to the lift. The six regions also match the recent study [10], so that our quantitative analysis can test previous hypotheses which were mostly based on the qualitative comparison. Besides, each partition included more than 20 characters to ensure sufficient information for inference. Partitioned analysis has been attempted on a much smaller dataset of stem birds under a Bayesian non-clock model to examine the branch-length variations across three body regions [16]. Since each branch length was a product of geological time duration and evolutionary rate, the two elements could not be distinguished without a time tree and clock assumption.

Here, we exploited the advantage of tip dating to infer both the divergence times and evolutionary rates while accounting for their uncertainties in a coherent Bayesian statistical framework. The technique was originally developed for analyses combining both morphological and molecular data [17–22] and has been termed as 'total-evidence dating'. It has also been productively applied to morphological data only [23–26] and we use 'tip dating' for that purpose. This approach has the essential strengths of incorporating various sources of information from the fossil record directly in the analysis, modelling the speciation process explicitly through a probabilistic model, allowing for parameter inference and model selection and using the state-of-the-art developments in Bayesian computation.

# 2. Material and methods

## 2.1. Tip dating

The revised morphological dataset contains 280 characters and 68 taxa. All characters are discrete (coded as 0, 1, etc.). One hundred and ninety-three of them are binary and the rest are multistate. Thirty-six characters were defined as ordered, which means instantaneous change is only allowed between adjacent states, while the rest of the characters were thus unordered. Only variable characters were coded, and the acquisition bias was corrected in the Mkv model [27]. Note that we did not remove any parsimony-uninformative characters as they are informative in model-based Bayesian inference.

In the Bayesian tip dating framework, we infer the posterior probability distribution of the model parameters, which combines the information from the morphological characters (likelihood) and the priors (including the distributions of the fossil ages and the other parameters in the tree model and clock model). For the likelihood, the Mkv model [27] was used for the character-state substitution with gamma rate variation across characters [28]. The prior for gamma shape ($\alpha$) was exponential (1), while the priors for the time tree and the relaxed clock model parameters are described in detail in the following sections. The method has been implemented in MrBayes v. 3.2.7 [29].

In order to root the tree properly and infer the evolutionary rates more reliably, we applied five topology constraints as Aves, Pygostylia, Ornithothoraces, Enantiornithes and Ornithuromorpha (figure 1), each of which forms a monophyletic clade.

The posterior distribution was estimated using Markov chain Monte Carlo (MCMC). We executed two independent runs with four chains (one cold and three hot) per run for 40 million iterations and sampled every 2000 iterations. The first 25% samples were discarded as burn-in for each run, and the remaining samples from the two runs were combined after checking consistency between runs.

## 2.2. Tree model

The fossilized birth-death (FBD) process [20,30–32] was used to model speciation, extinction, fossilization and sampling, which gave rise to the prior distribution of the time tree $\mathcal{T}$, including the topology ($\tau$) and branch lengths measured by Myr ($t_i$). The process starts at the root with two lineages sharing the same origin. Each lineage bifurcates with a constant rate $\lambda$ and goes extinct with a constant rate $\mu$. Concurrently, each lineage is sampled with a constant rate $\psi$ and is removed from the process upon sampling with probability $r$. Extant taxa are sampled with probability $\rho$. The explicit derivation of the probability density function was given in [32]. When the removal probability $0 \leq r < 1$, the sampled tree may contain fossil ancestors (i.e. fossils with sampled descendants), while setting $r = 1$ disables fossil ancestors (i.e. all fossils are at the tips).

The age of each fossil bird was assigned a uniform prior with lower and upper bounds from the corresponding stratigraphic range. The root age was assigned an offset exponential prior with the mean 169 Ma (slightly older than the first appearance datum of Dromaeosauridae) and minimum 153 Ma (slightly older than the first appearance datum of *Archaeopteryx*). For inference, we reparametrized the speciation, extinction and sampling rates and assigned priors as $d = \lambda - \mu - r\psi \sim$ exponential(100) with mean 0.01, $v = (\mu + r\psi)/\lambda$ and $s = \psi/(\mu + \psi) \sim$ uniform(0, 1). The sampling proportion of extant taxa (*Anas* and *Gallus*) was set to 0.0002, based on the number of described living bird species around 10 000 [33].

## 2.3. Clock model

We applied the independent gamma (white noise) relaxed clock model [34] to investigate evolutionary rate heterogeneity both along the tree and across the six anatomical regions (partitions). We did a new implementation of the model in MrBayes, which reparametrized the parameters aiming to articulate the relative rates. The original model applied to branch lengths was measured by distance. For branch $i$, the branch length $b_i$ is a product of geological time duration $t_i$ (in Myr) and clock rate $c_i$ (in unit of substitutions per character per Myr), and it is gamma distributed with the mean $t_i c$ and variance $t_i c \sigma$, where $c$ is the mean (base) clock rate and $\sigma$ is the variance parameter. Now, we define $c_i = cr_i$. Through variable transformation, $r_i$ is gamma distributed with the mean 1.0 and variance $\sigma/(t_i c)$. For multiple data partitions, we use independent variance parameter $\sigma_j$ for partition $j$ and $c_{ij} = cr_{ij}$. Thus, the clock model has $n + 1$ parameters ($c$, $\sigma_1, \ldots, \sigma_n$) for $n$ partitions. The relative rate $r_{ij}$ serves as a multiplier to the mean rate. Large deviation of $r_{ij}$ from 1.0 indicates severe heterogeneity of the morphological clock, while $r_{ij}$'s all being similar to 1.0 models a somewhat strict clock. The branch length (distance, in unit of substitutions per character) in the Mkv likelihood calculation, $b_{ij}$, is the product of time duration $t_i$ and clock rate $c_{ij}$ (i.e., $b_{ij} = t_i c_{ij} = t_i cr_{ij}$).

The prior used for the mean clock rate $c$ was gamma (2, 200) with the mean 0.01 and standard deviation 0.007, and that for $\sigma_j$ was exponential (10).

# 3. Results and discussion

There is no clear evidence for us to believe that all fossils are at the tips *a priori*; however, we encountered a severe mixing problem in the MCMC when allowing fossil ancestors while partitioning the

**4**

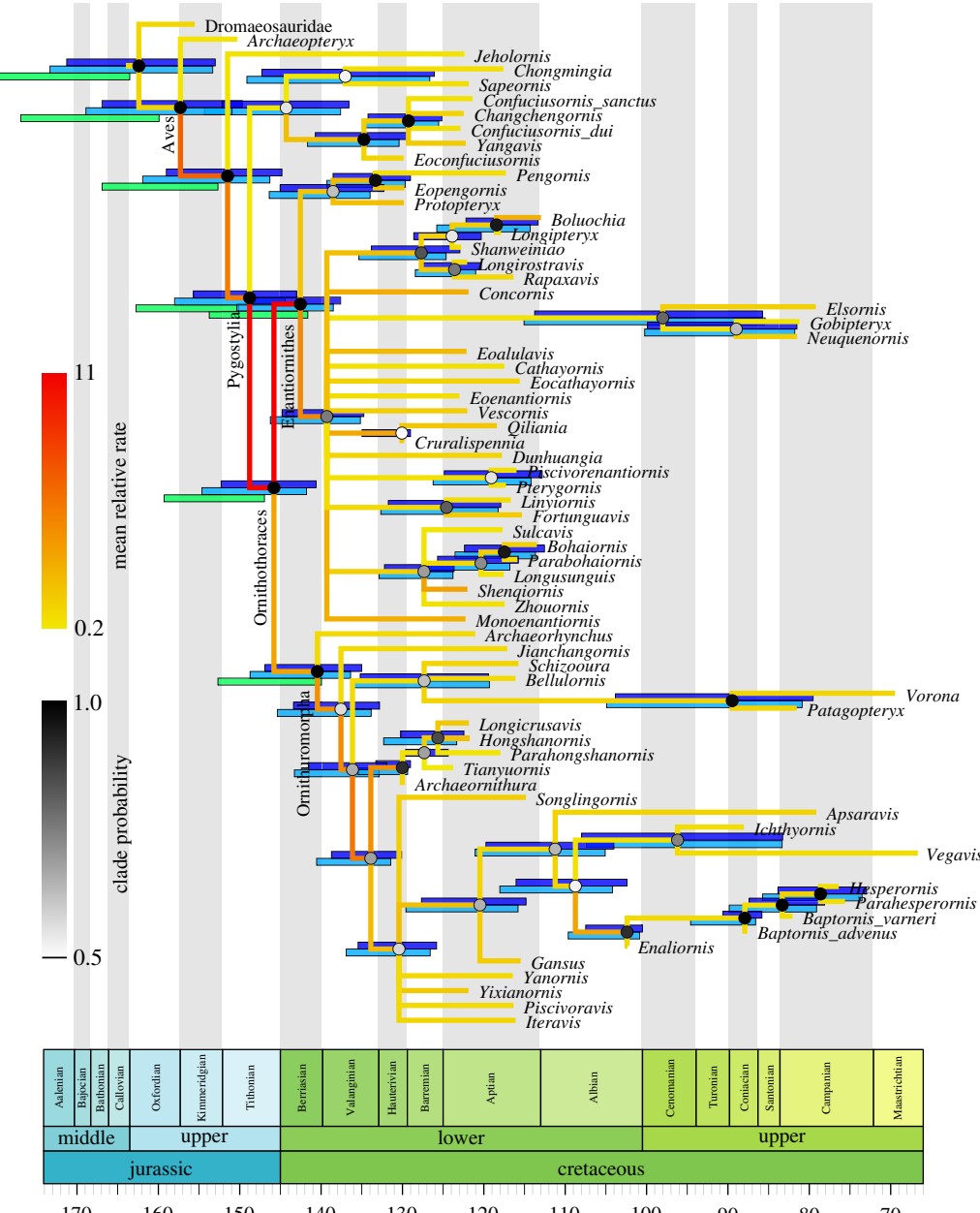

**Figure 1.** Dated phylogeny (time tree) of Mesozoic birds. The topology (majority-rule consensus tree) shown is inferred allowing fossil ancestors ($r = 0$) without partitioning the morphological characters. The node ages in the tree are the posterior medians and the shade of each circle represents the posterior probability of the corresponding clade. The colour of the branch represents the mean relative clock rate at that branch. The error bars on the top (in blue) at the internal nodes denote the 95% HPD intervals of age estimates. In comparison, the error bars below (if present, in cyan) denote the 95% HPD intervals when disallowing fossil ancestors ($r = 1$) under a single partition. Additionally, the error bars shown at the early avian diversifications (in green) are the 95% HPD intervals of age estimates when the characters are partitioned into six anatomical regions and disallowing fossil ancestors ($r = 1$). The two extant species (*Anas* and *Gallus*) were included in the analyses but not shown in the representation (as a sister clade of *Vegavis*).

morphological characters into six anatomical regions (see below). When treating the characters as a single partition on the other hand, we were able to achieve good mixing both allowing and disallowing fossil ancestors (setting $r = 0$ and $r = 1$, respectively). Thus, we first show the results without partitioning the data and compare the difference between with and without fossil ancestors, then we focus on the evolutionary rate heterogeneity when the data are partitioned (only under $r = 1$). In general, the parameter estimates are quite similar between fossil ancestors allowed and disallowed; the difference is more dramatic between the data were partitioned and unpartitioned.

**Table 1.** Posterior distributions (mean and 95% HPD interval) of model parameters.

| | single partition, $r = 0$ | single partition, $r = 1$ | six partitions, $r = 1$ |
|---|---|---|---|
| $\alpha$ | 1.81 (1.35, 2.31) | 1.82 (1.35, 2.31) | 1.80 (1.33, 2.27) |
| $d = \lambda - \mu - r\psi$ | 0.010 (0.00087, 0.022) | 0.010 (0.00075, 0.021) | 0.010 (0.00042, 0.020) |
| $v = (\mu + r\psi)/\lambda$ | 0.98 (0.94, 1.00) | 0.93 (0.81, 1.00) | 0.92 (0.79, 1.00) |
| $s = \psi/(\mu + \psi)$ | 0.028 (0.00047, 0.075) | 0.33 (0.00051, 0.90) | 0.35 (0.00059, 0.90) |
| $t_{mrca}$ | 162.56 (153.00, 171.26) | 164.17 (153.33, 173.34) | 172.61 (164.01, 180.81) |
| $c$ | 0.012 (0.0090, 0.015) | 0.011 (0.0083, 0.014) | 0.010 (0.0081, 0.012) |
| $\sigma$ or $\sigma_1$ | 0.058 (0.035, 0.084) | 0.058 (0.036, 0.084) | 0.070 (0.026, 0.12) |
| $\sigma_2$ | | | 0.069 (0.028, 0.12) |
| $\sigma_3$ | | | 0.057 (0.028, 0.090) |
| $\sigma_4$ | | | 0.053 (0.025, 0.087) |
| $\sigma_5$ | | | 0.042 (0.0017, 0.084) |
| $\sigma_6$ | | | 0.061 (0.029, 0.095) |

## 3.1. Single partition

The phylogeny estimated from tip dating allowing fossil ancestors ($r = 0$) is shown in figure 1. The tree is well resolved, with a few polytomies mainly nested within Enantiornithes that represent the uncertainty of taxonomic relationships. This topology agrees with previously published trees in the placements of the major clades [10,13,14]. The root age is estimated at 162.56 (153.00, 171.26) Ma (mean and 95% HPD interval, same for below; Table 1), which covers the fixed age of about 169 Ma [10] generated from the 'minimum branch length' [35] and 'equal' [36] methods. The posterior age of Dromaeosauridae is 154.90 (141.66, 167.69) Ma, mainly within the Late Jurassic, while the prior range expands the whole Cretaceous (66.0, 167.7). As the posterior mean relative rate at the branch of Dromaeosauridae is 1.08 (close to 1.0), the similarity of the morphological characters informs a short time span. The mean ages of the divergences of Pygostylia, Ornithothoraces and Enantiornithes are about 6–8 Myr older than the fixed ages in the previous study [10]. We emphasize that the age estimates using tip dating integrate all available sources of information, but the 'minimum branch length' and 'equal' methods only used the ages from adjacent nodes with ad hoc measures which are subjective and do not account for any uncertainties.

When disallowing fossil ancestors ($r = 1$), one concern that might arise is that some node ages might be overestimated due to forcing every tip to be the result of speciation. However, the difference from allowing fossil ancestors ($r = 0$) is really minor in our case. The node ages are only about 1–2 Myr older if not otherwise similar (figure 1). The topologies in these two cases are almost identical, except for two places—one is the placement of *Cruralispennia* which becomes unresolved in the big polytomy within Enantiornithes, the other is *Archaeornithura* which becomes a sibling of *Tianyuornis*. In fact, the proportion of fossil ancestors is 0.13 (0.06, 0.21), indicating that most fossils are indeed tip fossils. *Enaliornis* and *Archaeornithura* have the highest posterior probabilities of being ancestral (0.95 and 0.93, respectively).

The evolutionary rates are thus very similar in the two cases, and we only show the result under $r = 0$ (figure 1). The mean clock rate ($c$) is estimated at around 0.01 substitutions per character per Myr (i.e. approximately one character-state change per 100 million years) (table 1). The relative clock rate at each branch represents the deviation from the mean rate. We observe extremely high rates during early avian evolution (figure 1). The relative rates at the two branches from Aves to Pygostylia are 6.63 (0.97, 17.49) and 5.34 (0.01, 15.52), and accelerate to 10.89 (0.00, 32.5) at the branch subtending Ornithothoraces. High rates are also encountered along the early branches of Ornithuromorpha, then slow down substantially towards apical branches including the one leading to extant birds. Enantiornithes shows an even higher rate of 11.24 (1.49, 30.75) when it diverges from Ornithuromorpha in the Early Cretaceous, and the rates decrease dramatically in its later history. These observations concur with previous comparative studies that birds experienced substantial shifts in morphology in tandem with the dinosaur-bird transition [10,23,37].

The speciation rate and extinction rate are similar over the 100 million years evolution, indicated by small net diversification ($d$) and turnover ($v$) close to 1.0 (table 1). The current implementation does not allow negative net diversification ($\lambda < \mu$), so that mass extinction cannot be assessed at the moment. The sampling proportion ($s$) are distinct ($r = 0$ versus $r = 1$ in table 1) due to the fact that $\psi$ has different meanings in the fossil ancestor and non-ancestor analyses. In the fossil ancestor model, subsequent sampling can still happen after $\psi$ sampling of a lineage, so that $s$ can be inferred reliably; while in the non-ancestor model, $\psi$ has a similar effect as $\mu$ acting as removing lineages and they are partially identifiable, thus $s$ has a very large range between 0 and 1.

## 3.2. Six partitions

The evolutionary rates along the tree inferred above are averaged across all morphological characters. Further partitioning the data into six anatomical regions make it feasible for us to estimate refined evolutionary rates both along branches and across partitions. Different partitions have their own independent rate variations while sharing the same tree topology and geological time duration (i.e. single time tree $\mathcal{T}$). As mentioned above, the mixing was very poor if fossil ancestors were allowed ($r = 0$). Independent runs produced inconstant rate estimates and trees with different proportions of sampled ancestors, while fine-tuning the proposals, adding more heated chains and prolonging the runs did not improve the outcome much. The difficulty was probably due to limited data in each partition interfering with inefficient proposals implemented for updating the status of fossils. A reversible-jump MCMC (rjMCMC) algorithm [38] was used for each fossil to shift between being an ancestor and a tip [20,31,32]. The algorithm generates a branch length uniformly between the age of the fossil and the age of its parent when proposing an ancestral fossil to become a tip and discarding the branch length in the reversed move. Further development of more efficient proposals is required. At the moment, we only show the result disallowing fossil ancestors ($r = 1$) which simplifies the tree structure with no need for rjMCMC. In this case, the major clades inferred in the tree are unchanged, although a few taxa with large uncertainties shuffle a bit (electronic supplementary material, figures S1–S6). Compared with the node ages estimated under a single partition, those at the early avian diversifications are slightly older (figure 1, see also table 1) while the younger ages become more similar. The age differences are more dramatic whether the data are partitioned than whether fossil ancestors are allowed.

The rates of morphological character changes are quite heterogeneous among the six regions during early avian evolution (figure 2), although the mean rate ($c$) estimated is almost identical as before (table 1). The branch subtending Pygostylia shows an extremely high rate in the axial skeleton (figure 2; electronic supplementary material, figure S2), which is one order of magnitude higher than in the rest of the five regions. Clearly, the high rate observed here indicates extensive morphological changes in the vertebral column, and the most recognizable change is that a long tail consisting of over 20 caudal vertebrae in *Archaeopteryx* and *Jeholornis* is replaced by a short element called pygostyle, which is formed by the fusion of several caudal most vertebrae [4]. The transition from long to short tail is functionally important for the evolution of powered flight in birds: a short tail could forward the gravitational centre, and with attached feathers become indispensable for the avian flight apparatus [39]. The branches subtending Ornithothoraces and Enantiornithes show very rapid morphological changes in the pectoral girdle and sternum and moderately high rates in the forelimb (figure 2; electronic supplementary material, figures S3 and S4). In comparison, the corresponding rates in the other regions are close to 1.0 with a slight variation. These results suggest that most of the changes related to the shoulder and forelimb are close to the origin of Ornithothoraces, for example, the presence of an ossified sternum with a keel (attachment for the major flight muscle in modern birds) and further elongated forelimb [9], contributing to the refinement of flight capability. The rapid morphological changes towards Enantiornithes correspond to their unique shoulder morphology relative to the ancestor Ornithothoraces; for instance, enantiornithines have a sternum with a caudally restricted keel and an elongate acromion of the scapula, both of which are the major components of the flight apparatus in birds [4,40]. The previous morphometric study suggested that Enantiornithes have a different flight style compared with other Mesozoic birds in terms of limb proportion [41]. The rest of the relatively slow and even rates suggest that there is no dominant selective pressure in favouring of modifications in the skull and pelvis.

When we compared the evolutionary rates between Enantiornithes and Ornithuromorpha across the six regions by summarizing the mean relative rates within each clade (excluding *Anas*, *Gallus* and their common ancestral branch), the differences are not striking, with medians uniformly close to 1.0 (figure 3). However, significantly high rates (outliers) are detected along some early diverging branches within

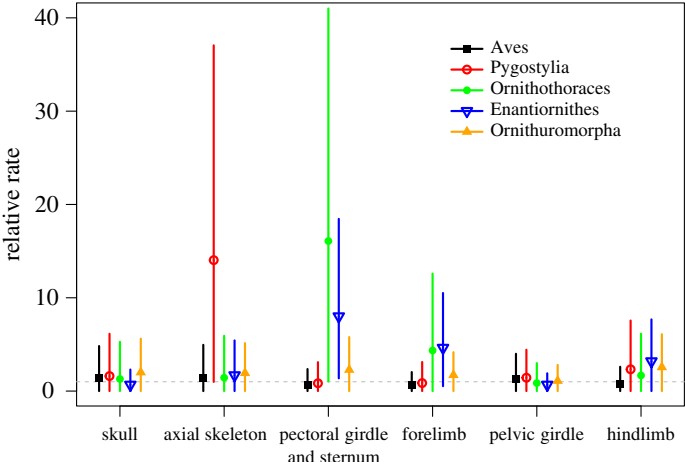

**Figure 2.** Posterior estimates of the relative clock rates along the branches subtending the major transitions of early avian evolution for six anatomical regions of the bird body. The dot and error bar denote the mean and 95% HPD interval for each estimate, respectively. The horizontal dashed line indicates the mean relative rate of 1.0 in the relaxed model.

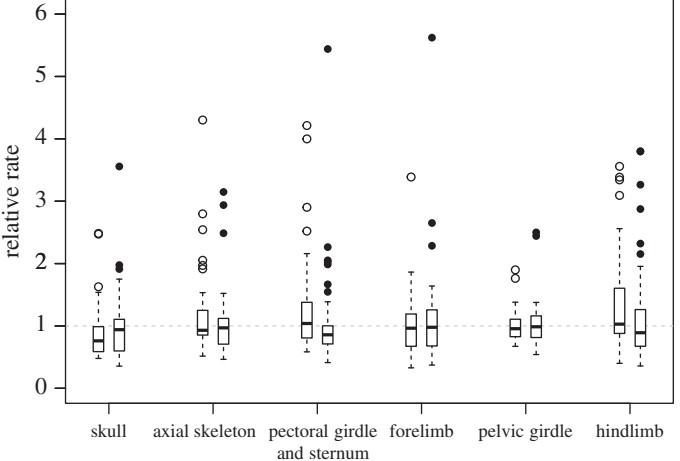

**Figure 3.** Boxplots summarizing the mean relative clock rates across branches in the Enantiornithes clade (left) and Ornithuromorpha clade (right), respectively, for the six anatomical regions. The box denotes the first, second (median) and third quartiles while the dots are the outliers. The horizontal dashed line indicates the mean relative rate of 1.0 in the relaxed clock model.

these two clades, and a slowdown in more crownward branches (figure 3; electronic supplementary material, figures S1–S6). This suggests that morphological changes are rapid in early diversifications, and the process slows down subsequently due to saturated ecological niches for these two avian groups, as proposed in the previous study [10].

To test the robustness of age estimates to the root age prior and the impact on evolutionary rates in consequence, we halved and doubled the range between the mean and minimal in the original prior, that is, using offset-exp(153, 161) and offset-exp(153, 185) priors for the root age. This comparison showed slightly varied posterior age estimates deep in the tree. For the root age in particular, the estimates are 170.83 (161.33, 179.78) and 174.31 (164.34, 184.77) under the smaller and larger prior mean, respectively. Besides, we also changed the prior mean for the mean clock rate by 10-fold, that is, using gamma(2, 20) with mean 0.1 and gamma(2, 2000) with mean 0.001 for $c$. The posterior estimate of $c$ is merely changed; subsequently, the time estimates are not influenced much. Such robustness tests confirm that the fossil ages provide good information to distinguish the times and rates in the inference, and these priors are less influential. Thus, the conclusion of evolutionary rate heterogeneity above is not changed as the age differences are minor compared with the significantly high rates which are usually one order of magnitude higher than 1.0.

Another concern related to the accuracy of age estimates might be the non-uniform fossil record both in its geographical and stratigraphic distribution. Since the focus of our study is on the deep divergence

times and the evolutionary rates, the rich fossil record in the Early Cretaceous provided a great deal of information to produce reliable estimates deep in the tree. For the age estimates in the Late Cretaceous and later, biases might be severe due to the limited fossil record. In particular, the divergence time of *Anas* and *Gallus* seems too young compared with other studies focusing on crown birds [3] (approx. 10 versus 55 Ma). The morphological characters of *Anas* and *Gallus* only provide information about their evolutionary distance, meaning that the same distance can result from either fast rate evolved in a short time or slow rate in a long time. To distinguish the time and rate, we further added a node calibration on the root of *Anas* and *Gallus* and set a normal distribution prior with the mean 55 Ma and standard deviation 5 Myr. The posterior age shifts to 44.38 (33.82, 54.93) Ma in the Eocene, more in line with previous studies.

The gamma shape parameter ($\alpha$) of character rate variation is larger than 1.0 (table 1), indicating that the evolutionary rates are fairly homogeneous across characters. Note that this gamma distribution models rate variation across characters and is independent of the gamma distributions for rate variation across branches in the relaxed clock model.

Partitioning the data is a common practice in molecular phylogenetic analyses [42,43]. The different partitions may correspond to different genes and may also correspond to different codon positions in a protein-coding gene. On the other hand, morphological characters are typically treated as a single partition unless sufficient characters are available [21] or simple model assumptions are made [16]. This study revealed the heterogeneous evolutionary dynamics modelled by independent rate parameters in the six partitions, which cannot be discovered under a single partition. Nevertheless, the limited amount of morphological characters, even if the largest for the group that has ever been assembled, restricts the method to produce more precise time and rate estimates. For example, some estimates showed very large variances (figure 2, widths of the error bars). Further efforts to code more characters would refine the resolution. The stratigraphic age uncertainty represented as uniform distributions also contributes to the variances, but ignoring such uncertainty can lead to biased age estimates under the FBD process [44]. In total-evidence dating, a large amount of molecular data can typically produce reliable backbone relationships of extant taxa. But for the Mesozoic birds under study, the extant taxa would form a monophyletic group in the Paleogene and thus would not contribute much to the estimation of divergence times and evolutionary rates in the stem group. Thus, abundant high-quality morphological characters and accurate stratigraphic ages of the fossils are critical.

# 4. Conclusion

The Bayesian tip dating approach implemented in MrBayes is a powerful and flexible tool to simultaneously estimate the tree topology, divergence times, evolutionary rates and the other parameters of interest while accounting for their uncertainties. In the priors, we are able to incorporate the uncertainty of each fossil age, model the speciation, extinction, fossilization and sampling process explicitly, and take advantage of the relaxed clock model to investigate rate variation cross branches and partitions. It is feasible to integrate all available sources of information in the analysis, rather than discarding certain information or uncertainties in the parsimony and stepwise approach. Although the focal species are Mesozoic birds in this study, tip dating is a general framework applicable to a wide range of taxonomic groups with potential future extensions to the theoretical model and practical implementation.

Data accessibility. Data available from the Dryad Digital Repository: https://doi.org/10.5061/dryad.20c2r58 [45].

Authors' contributions. C.Z. designed the study and performed the analyses. M.W. revised the dataset. C.Z. and M.W. wrote and revised the manuscript. Both authors gave final approval for publication.

Competing interests. We declare we have no competing interests.

Funding. This research is supported by the 100 Young Talents Program of Chinese Academy of Sciences and the Strategic Priority Research Program of Chinese Academy of Sciences (XDB26000000), both to C.Z. M.W. is supported by the National Natural Science Foundation of China (41722202).

Acknowledgements. We thank the anonymous reviewers for constructive suggestions for improving the manuscript.

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
