## [Reviewer comments · Royal Society Open Science]

Review History

RSOS-182062.R0 (Original submission)

Review form: Reviewer 1

Is the manuscript scientifically sound in its present form?

Yes

Are the interpretations and conclusions justified by the results?

Yes

Is the language acceptable?

Yes

Is it clear how to access all supporting data?

Yes

Do you have any ethical concerns with this paper?

No

Have you any concerns about statistical analyses in this paper?

No

Recommendation?

Accept with minor revision (please list in comments)

Comments to the Author(s)

To the editor and the authors,

I appreciate the chance to review the manuscript “Bayesian Tip Dating Reveals Heterogenous Morphological Clocks in Mesozoic Birds”. Firstly, I’d like to apologize for being late by a week to submit my review – it was a busy month, teaching-wise for me. I found the manuscript to be quite clear and understandable, although I admit that I am overly familiar with the methods described and thus maybe not quite so liable to notice if it is opaque to the general reader. I think the questions and the findings will be of broad interest to the paleontological and systematic community – in fact before I had been asked to review it, I had already downloaded the manuscript from bioRxiv, and put it into my ‘read as soon as I have time to spare’ pile. Overall, I think the paper is mainly ready for acceptance, but there are some issues that could be better handled in how they are discussed and presented by the authors. I list my concerns regarding these areas below, however I don’t foresee these issues requiring much additional work from the authors. Thus, I recommend acceptance.

So, here’s the major issues I had:

The authors are quite careful to spend a great deal of time addressing the inability to get sampled-ancestor (SA) analyses to converge under the heterogenous morphological model. This is good, because if you know much about tip-dating and the differences between the SA and non-SA models, this is very concerning, and it is good to see them address much time to it. However, it is unclear to me how much they tried – if the topology they get is fairly similar across analyses that were both SA and non-SA, and both homogenous-rate and heterogenous-rate morphological models, did they try constraining the topology considerably to all points of agreement, and then doing the SA heterogenous-morph analyses? Based on my reading, it doesn’t seem like that route was explored. I don’t really think if they’d been able to get an SA analysis to converge that it would have made a difference in their findings, but I’d like to at least see discussion from the authors about how future workers might tackle lack of convergence.

Why these six anatomical regions? Why not six other regions, or less, or more? This aspect of the author’s decisions could be much better addressed.

I’m curious about the ‘minimum branch length’ method mentioned on line 154, as it cites Wang & Lloyd. Is this the minimum branch length method devised by Laurin, as implemented in paleotree? There should probably be references to the method, rather than just to a previous analysis by one of the authors. And why is only minimum branch length a posteriori time-scaling discussed – did the outcome not differ much from the ‘equal’ method a posteriori time-scaling in Wang & Lloyd (it looks like both were applied in Wang & Lloyd)?

Several times in the paper a number is given, followed by two other number in parentheses, separated by a comma. Is that a range? A quantile of some sort?

Your SA analyses has an order of magnitude higher probability of fossil taxa being sampled than your non-SA analyses, looking at your Table 1. Any thoughts why?

There’s a link to a Dryad repository in the manuscript, but I do not seem to be able to access any

supplemental materials there. My comments would not be greatly changed by having those materials to access, though – these issues should be addressed in the main text.

Minor Comments, by line number:

Lines 182-183: That's very interesting – is there any reason those particular taxa have high posterior support of being sampled ancestors? (Don't remove this line if there isn't – I'm just curious.)

Lines 196: 'birds underwent a large scale of diversification' – this is worded a bit oddly; 'diversification' usually means speciation or the change in taxonomic richness/diversity in the macroevolutionary literature, but I think the authors mean more a change in morphological evolution and not the accrual of lineages, so maybe '...birds experienced substantial shifts in morphology in tandem with...' instead?

Lines 207-216: I'm very confused here. What is an rjMCMC? I wouldn't characterize sampled-ancestor-moves as being a reversible-jump algorithm, and I don't characterize how the authors described their heterogenous morphology model as a reversible jump, so what is a reversible jump? This is the only place where reversible-jump MCMC is described in the paper at all, to make it even more confusing.

Review form: Reviewer 2

Is the manuscript scientifically sound in its present form?

Yes

Are the interpretations and conclusions justified by the results?

Yes

Is the language acceptable?

Yes

Is it clear how to access all supporting data?

Yes

Do you have any ethical concerns with this paper?

No

Have you any concerns about statistical analyses in this paper?

No

Recommendation?

Major revision is needed (please make suggestions in comments)

Comments to the Author(s)

This paper analyzes a morphological matrix of bird fossils using the mrBayes program to estimate the species phylogeny, divergence times, and evolutionary rates. It is an update and re-analysis of a previous study using the parsimony method. I think the questions addressed are interesting and the manuscript is clearly written. I support its publication after a revision.

1. Include a brief description/summary of the data matrix. Mention that the characters are discrete or discretized, how many are binary and how many are multistate, and how many are

ordered. I assume that the characters are variable across the taxa. Are parsimony-noninformative characters removed, and if so, how is the assertion bias dealt with in the program, etc.

2. There should be a robustness analysis, to examine the posterior sensitivity to the priors. The model is extremely parameter-rich so that it will be hard to do a comprehensive robustness analysis, but the impact of some of the important parameters should be examined. For example, I wonder how the posterior times change when change the prior mean for the mean clock rate c by 10 folds. The $U(0, 1)$ priors on the transformed birth-death parameters look too informative and restrictive. The posterior for ν is close to 1, suggesting that the prior truncated values higher than 1.

3. The authors described the advantages of the Bayesian approach and tip dating, such as being able to integrate information from different sources. Such discussions are OK in theory, but in practice there are many caveats in such an analysis, given that the data involve a lot of uncertainties and the model has many parameters for which there is little information in the data. Basically we are asking a lot of difficult questions from very limited data (even if the dataset is the largest for the group that has ever been assembled). I would like the authors to discuss some of the caveats and take a more measured tone in the writing. For example, the use of the rock strata to form uniform bounds on the ages of the fossils involve a lot of uncertainties. The posterior estimates of times (and rates) have large intervals. This partly reflects the uncertainties in the analysis but limits the utility of the estimates.

4. No molecular data are used in this study. Perhaps briefly comment on the advantages and disadvantages of combining molecular and morphological data in such an analysis. When you include modern species, presumably morphological characters from modern species could be used together from morphological characters based on fossils?

5. Personally I do not think partitioning is necessary: it makes the model too parameter rich to be barely usable. First, there are only a few hundred characters. Second the need to correct for the assertion bias means further loss of information. And thirdly, you already have a gamma model to accommodate variable rates among characters. I suggest that you look at the rate estimates for the characters and branches under the one-partition model, and average over the six regions, and confirm that the trend is the same as you see from the six-partition analysis.

6. p.7 line 134. Why is the variance for the gamma branch rate inversely proportional to the time duration of the branch (t_i). This does not make biological sense. In the geometric Brownian motion model of Thorne et al., the variance for the log rate is proportional to the time duration, so that the longer the branch the more the rate will drift.

7. The ms. will benefit from some editing.

Decision letter (RSOS-182062.R0)

30-May-2019

Dear Dr Zhang,

The editors assigned to your paper ("Bayesian tip dating reveals heterogeneous morphological clocks in Mesozoic birds") have now received comments from reviewers. We would like you to revise your paper in accordance with the referee and Associate Editor suggestions which can be found below (not including confidential reports to the Editor). Please note this decision does not guarantee eventual acceptance.

Please submit a copy of your revised paper before 22-Jun-2019. Please note that the revision deadline will expire at 00.00am on this date. If we do not hear from you within this time then it will be assumed that the paper has been withdrawn. In exceptional circumstances, extensions

may be possible if agreed with the Editorial Office in advance. We do not allow multiple rounds of revision so we urge you to make every effort to fully address all of the comments at this stage. If deemed necessary by the Editors, your manuscript will be sent back to one or more of the original reviewers for assessment. If the original reviewers are not available, we may invite new reviewers.

- Data accessibility

<http://datadryad.org/submit?journalID=RSOS&manu=RSOS-182062>

- Competing interests

- Authors' contributions

- Acknowledgements

- Funding statement

Kind regards,

Alice Power

on behalf of Professor Matthew Collins (Associate Editor) and Jon Blundy (Subject Editor)

Comments to Author:

Reviewers' Comments to Author:

Reviewer: 1

Comments to the Author(s)

To the editor and the authors,

I appreciate the chance to review the manuscript "Bayesian Tip Dating Reveals Heterogenous Morphological Clocks in Mesozoic Birds". Firstly, I'd like to apologize for being late by a week to submit my review - it was a busy month, teaching-wise for me. I found the manuscript to be quite clear and understandable, although I admit that I am overly familiar with the methods described and thus maybe not quite so liable to notice if it is opaque to the general reader. I think the questions and the findings will be of broad interest to the paleontological and systematic community - in fact before I had been asked to review it, I had already downloaded the manuscript from bioRxiv, and put it into my 'read as soon as I have time to spare' pile. Overall, I think the paper is mainly ready for acceptance, but there are some issues that could be better handled in how they are discussed and presented by the authors. I list my concerns regarding these areas below, however I don't foresee these issues requiring much additional work from the authors. Thus, I recommend acceptance.

So, here's the major issues I had:

The authors are quite careful to spend a great deal of time addressing the inability to get sampled-ancestor (SA) analyses to converge under the heterogeneous morphological model. This is good, because if you know much about tip-dating and the differences between the SA and non-SA models, this is very concerning, and it is good to see them address much time to it. However, it is unclear to me how much they tried – if the topology they get is fairly similar across analyses that were both SA and non-SA, and both homogeneous-rate and heterogeneous-rate morphological models, did they try constraining the topology considerably to all points of agreement, and then doing the SA heterogeneous-morph analyses? Based on my reading, it doesn't seem like that route was explored. I don't really think if they'd been able to get an SA analysis to converge that it would have made a difference in their findings, but I'd like to at least see discussion from the authors about how future workers might tackle lack of convergence.

Why these six anatomical regions? Why not six other regions, or less, or more? This aspect of the author's decisions could be much better addressed.

I'm curious about the 'minimum branch length' method mentioned on line 154, as it cites Wang & Lloyd. Is this the minimum branch length method devised by Laurin, as implemented in paleotree? There should probably be references to the method, rather than just to a previous analysis by one of the authors. And why is only minimum branch length a posteriori time-scaling discussed – did the outcome not differ much from the 'equal' method a posteriori time-scaling in Wang & Lloyd (it looks like both were applied in Wang & Lloyd)?

Several times in the paper a number is given, followed by two other number in parentheses, separated by a comma. Is that a range? A quantile of some sort?

Your SA analyses has an order of magnitude higher probability of fossil taxa being sampled than your non-SA analyses, looking at your Table 1. Any thoughts why?

There's a link to a Dryad repository in the manuscript, but I do not seem to be able to access any supplemental materials there. My comments would not be greatly changed by having those materials to access, though – these issues should be addressed in the main text.

Minor Comments, by line number:

Lines 182-183: That's very interesting – is there any reason those particular taxa have high posterior support of being sampled ancestors? (Don't remove this line if there isn't – I'm just curious.)

Lines 196: 'birds underwent a large scale of diversification' – this is worded a bit oddly; 'diversification' usually means speciation or the change in taxonomic richness/diversity in the macroevolutionary literature, but I think the authors mean more a change in morphological evolution and not the accrual of lineages, so maybe '...birds experienced substantial shifts in morphology in tandem with...' instead?

Lines 207-216: I'm very confused here. What is an rjMCMC? I wouldn't characterize sampled-ancestor-moves as being a reversible-jump algorithm, and I don't characterize how the authors described their heterogeneous morphology model as a reversible jump, so what is a reversible jump? This is the only place where reversible-jump MCMC is described in the paper at all, to make it even more confusing.

Reviewer: 2

Comments to the Author(s)

This paper analyzes a morphological matrix of bird fossils using the mrBayes program to estimate the species phylogeny, divergence times, and evolutionary rates. It is an update and re-analysis of a previous study using the parsimony method. I think the questions addressed are interesting and the manuscript is clearly written. I support its publication after a revision.

1. Include a brief description/summary of the data matrix. Mention that the characters are discrete or discretized, how many are binary and how many are multistate, and how many are ordered. I assume that the characters are variable across the taxa. Are parsimony-noninformative characters removed, and if so, how is the ascertainment bias dealt with in the program, etc.
2. There should be a robustness analysis, to examine the posterior sensitivity to the priors. The model is extremely parameter-rich so that it will be hard to do a comprehensive robustness analysis, but the impact of some of the important parameters should be examined. For example, I wonder how the posterior times change when change the prior mean for the mean clock rate c by 10 folds. The $U(0, 1)$ priors on the transformed birth-death parameters look too informative and restrictive. The posterior for ν is close to 1, suggesting that the prior truncated values higher than 1.
3. The authors described the advantages of the Bayesian approach and tip dating, such as being able to integrate information from different sources. Such discussions are OK in theory, but in practice there are many caveats in such an analysis, given that the data involve a lot of uncertainties and the model has many parameters for which there is little information in the data. Basically we are asking a lot of difficult questions from very limited data (even if the dataset is the largest for the group that has ever been assembled). I would like the authors to discuss some of the caveats and take a more measured tone in the writing. For example, the use of the rock strata to form uniform bounds on the ages of the fossils involve a lot of uncertainties. The posterior estimates of times (and rates) have large intervals. This partly reflects the uncertainties in the analysis but limits the utility of the estimates.
4. No molecular data are used in this study. Perhaps briefly comment on the advantages and disadvantages of combining molecular and morphological data in such an analysis. When you include modern species, presumably morphological characters from modern species could be used together from morphological characters based on fossils?
5. Personally I do not think partitioning is necessary: it makes the model too parameter rich to be barely usable. First, there are only a few hundred characters. Second the need to correct for the ascertainment bias means further loss of information. And thirdly, you already have a gamma model to accommodate variable rates among characters. I suggest that you look at the rate estimates for the characters and branches under the one-partition model, and average over the six regions, and confirm that the trend is the same as you see from the six-partition analysis.
6. p.7 line 134. Why is the variance for the gamma branch rate inversely proportional to the time duration of the branch (t_i). This does not make biological sense. In the geometric Brownian motion model of Thorne et al., the variance for the log rate is proportional to the time duration, so that the longer the branch the more the rate will drift.
7. The ms. will benefit from some editing.

Author's Response to Decision Letter for (RSOS-182062.R0)

See Appendix A.

Decision letter (RSOS-182062.R1)

21-Jun-2019

Dear Dr Zhang,

I am pleased to inform you that your manuscript entitled "Bayesian tip dating reveals heterogeneous morphological clocks in Mesozoic birds" is now accepted for publication in Royal Society Open Science.

When reading through the proofs, please make the following minor editorial changes to the text, in accordance with the recommendations of the Associate Editor:

1. 'time tree' should be either hyphenated or concatenated.
2. 'root-age' should be hyphenated

These are very minor changes, that can easily be accomplished at the proof stage.

on behalf of Professor Matthew Collins (Associate Editor) and Jon Blundy (Subject Editor)
openscience@royalsociety.org

Appendix A

中国科学院古脊椎动物与古人类研究所

INSTITUTE OF VERTEBRATE PALEONTOLOGY AND PALEOANTHROPOLOGY

中国科学院

CHINESE ACADEMY OF SCIENCES

Key Laboratory of Vertebrate Evolution and Human Origins of Chinese Academy of Sciences

Chi Zhang

Associate Professor

142 XiZhiMenWai Street, Beijing 100044, China

Email: zhangchi@ivpp.ac.cn

Phone: +86 10 8836 0750

Re: Revision of " Bayesian tip dating reveals heterogeneous morphological clocks in Mesozoic birds " by Zhang, Chi and Wang, Min.

Thank you for the referee comments on our manuscript. We have revised the manuscript according to these suggestions as listed below. We submit a clean version together with one with track changes enabled. We hope that our manuscript is ready for publication in *Royal Society Open Science*. Thank you very much for your time to handle this manuscript.

Yours sincerely,

Chi Zhang and Min Wang

Beijing, June 3, 2019

30-May-2019

Dear Dr Zhang,

The editors assigned to your paper ("Bayesian tip dating reveals heterogeneous morphological clocks in Mesozoic birds") have now received comments from reviewers. We would like you to revise your paper in accordance with the referee and Associate Editor suggestions which can be found below (not including confidential reports to the Editor). Please note this decision does not guarantee eventual acceptance.

Please submit a copy of your revised paper before 22-Jun-2019. Please note that the revision deadline will expire at 00.00am on this date. If we do not hear from you within this time then it will be assumed that the paper has been withdrawn. In exceptional circumstances, extensions may be possible if agreed with the Editorial Office in advance. We do not allow multiple rounds of revision so we urge you to make every effort to fully address all of the comments at this stage. If deemed necessary by the Editors, your manuscript will be sent back to one or more of the original reviewers for assessment. If the original reviewers are not available, we may invite new reviewers.

To revise your manuscript, log into <http://mc.manuscriptcentral.com/rsos> and enter your

Author Centre, where you will find your manuscript title listed under "Manuscripts with Decisions." Under "Actions," click on "Create a Revision." Your manuscript number has been appended to denote a revision. Revise your manuscript and upload a new version through your Author Centre.

-- This is not applicable to our study.

- Data accessibility

If you wish to submit your supporting data or code to Dryad (<http://datadryad.org/>), or modify your current submission to dryad, please use the following link:
<http://datadryad.org/submit?journalID=RSOS&manu=RSOS-182062>

-- Data accessibility section included with Dryad DOI:
<https://doi.org/10.5061/dryad.20c2r58>.

- Competing interests

-- included

- Authors' contributions

-- included

- Acknowledgements

-- included

- Funding statement

-- included

Kind regards,
Alice Power
Royal Society Open Science
openscience@royalsociety.org

on behalf of Professor Matthew Collins (Associate Editor) and Jon Blundy (Subject Editor)
openscience@royalsociety.org

Comments to Author:

Reviewers' Comments to Author:

Reviewer: 1

Comments to the Author(s)

To the editor and the authors,

I appreciate the chance to review the manuscript "Bayesian Tip Dating Reveals Heterogenous Morphological Clocks in Mesozoic Birds". Firstly, I'd like to apologize for being late by a week to submit my review – it was a busy month, teaching-wise for me. I found the manuscript to be quite clear and understandable, although I admit that I am overly familiar with the methods described and thus maybe not quite so liable to notice if it is opaque to the general reader. I think the questions and the findings will be of broad interest to the paleontological and systematic community – in fact before I had been asked to review it, I had already downloaded the manuscript from bioRxiv, and put it into my 'read as soon as I have time to spare' pile. Overall, I think the paper is mainly ready for acceptance, but there are some issues that could be better handled in how they are discussed and presented by the authors. I list my concerns regarding these areas below, however I don't foresee these issues requiring much additional work from the authors. Thus, I recommend acceptance.

-- Thank you for the affirmative recommendation.

So, here's the major issues I had:

The authors are quite careful to spend a great deal of time addressing the inability to get sampled-ancestor (SA) analyses to converge under the heterogenous morphological model. This is good, because if you know much about tip-dating and the differences between the SA and non-SA models, this is very concerning, and it is good to see them address much time to it. However, it is unclear to me how much they tried – if the topology they get is fairly similar across analyses that were both SA and non-SA, and both homogenous-rate and heterogenous-rate morphological models, did they try constraining the topology considerably to all points of agreement, and then doing the SA heterogenous-morph analyses? Based on my reading, it doesn't seem like that route was explored. I don't really think if they'd been able to get an SA analysis to converge that it would have made a difference in their findings, but I'd like to at least see discussion from the authors about how future workers might tackle lack of convergence.

-- We indeed tried several aspects to attempt better convergence under the heterogenous-rate model, including finetuning the proposals' tuning parameters and prolonging the runs. We also tried adding more heated chains. The outcome doesn't improve much. We spotted that the difficulty was mainly in updating the status of fossils. Some runs inferred more fossil ancestors than other runs, this likely caused inconsistent estimate of some other parameters. The program implemented a reversible jump MCMC algorithm for a fossil to "jump" between being a tip and an ancestor. It seems a more efficient proposal is needed to get out of the mire in this case. We have put more details to discuss this issue, and foresee this as a future work to tackle this problem. (these are added to the first paragraph in section (b) Six partitions)

It is not trivial to add more topological constraints as there are a lot of uncertainties about the placement of the fossils, especially in the Enantiornithines clade. We ended up enforcing only five major clades in our analyses. Nevertheless, it might be a good option to try for another dataset that has good “backbone” taxa relationships. Typically a dataset combining both morphological and molecular partitions would provide such an example, but our data do not apply. This is also reflected in the discussion. See also response to reviewer 2.

Why these six anatomical regions? Why not six other regions, or less, or more? This aspect of the author’s decisions could be much better addressed.

-- We have addressed this question in the introduction.

I’m curious about the ‘minimum branch length’ method mentioned on line 154, as it cites Wang & Lloyd. Is this the minimum branch length method devised by Laurin, as implemented in paleotree? There should probably be references to the method, rather than just to a previous analysis by one of the authors. And why is only minimum branch length a posteriori time-scaling discussed – did the outcome not differ much from the ‘equal’ method a posteriori time-scaling in Wang & Lloyd (it looks like both were applied in Wang & Lloyd)?

-- Both ‘mbl’ and ‘equal’ methods give similar outcomes. We have added references to the original methods. We note in the result that both methods use ad hoc measures and inform node age only from the adjacent nodes, by contrast, tip-dating infers node ages from all the data and is more objective and accounts for uncertainties. (in the first paragraph of section (a) Single partition)

Several times in the paper a number is given, followed by two other number in parentheses, separated by a comma. Is that a range? A quantile of some sort?

-- The explanation was given when the numbers first appeared: “The root age is estimated at 162.56 (153.00, 171.26) Ma (mean and 95% HPD interval, same for below. Table 1)”. It is also included in the figure legends.

Your SA analyses has an order of magnitude higher probability of fossil taxa being sampled than your non-SA analyses, looking at your Table 1. Any thoughts why?

-- The sampling proportion (s) are distinct due to the fact that ψ has different meanings in the fossil ancestor and non-ancestor analyses. In the fossil ancestor model, subsequent sampling can still happen after ψ -sampling of a lineage, so that s can be inferred reliably; while in the non-ancestor model, ψ has a similar effect as μ acting as removing lineages and they are not distinguishable, thus s has a very large range between 0 and 1. This is added to the result (in the end of section (a) Single partition).

There’s a link to a Dryad repository in the manuscript, but I do not seem to be able to access any supplemental materials there. My comments would not be greatly changed by having those materials to access, though – these issues should be addressed in the main text.

-- We have deposited the raw data in Dryad, and also in the supplementary information submitted this time. Once the manuscript is accepted and published, the Dryad link should become active and accessible by everyone.

Minor Comments, by line number:

Lines 182-183: That's very interesting – is there any reason those particular taxa have high posterior support of being sampled ancestors? (Don't remove this line if there isn't – I'm just curious.)

-- Based on the data (including morphological characters and ages of the fossils) and the model (prior), the MCMC algorithm updates the fossil states (tip or ancestor) and these two fossils ends up being ancestral with high posterior support. From the inference aspect, this simply means that the outcome has higher posterior than those being tips. Biologically, it is harder to justify. They might be true ancestors, but without more solid evidence, we just simply show the inference result.

Lines 196: 'birds underwent a large scale of diversification' – this is worded a bit oddly; 'diversification' usually means speciation or the change in taxonomic richness/diversity in the macroevolutionary literature, but I think the authors mean more a change in morphological evolution and not the accrual of lineages, so maybe '...birds experienced substantial shifts in morphology in tandem with...' instead?

-- The sentence is reworded as suggested.

Lines 207-216: I'm very confused here. What is an rjMCMC? I wouldn't characterize sampled-ancestor-moves as being a reversible-jump algorithm, and I don't characterize how the authors described their heterogenous morphology model as a reversible jump, so what is a reversible jump? This is the only place where reversible-jump MCMC is described in the paper at all, to make it even more confusing.

-- The move is simply described as rjMCMC in Heath et al 2014 and named "leaf to sampled ancestor jump" in Gavryushkina et al 2014 and "add and delete branch moves" in Zhang et al 2016. These papers are now cited together with Green's paper introducing the algorithm. Some details of the move are also provided.

The name is indeed confusing but it is inherited from the original paper and has been widely used in the literature. It is a rjMCMC move because it is a trans-dimensional MCMC. To change a fossil from an ancestor to a tip, a new branch is added to the fossil, which increases the dimension of the parameters (branches) by one. The reverse move decreases the dimension by one.

Reviewer: 2

Comments to the Author(s)

This paper analyzes a morphological matrix of bird fossils using the mrBayes program to estimate the species phylogeny, divergence times, and evolutionary rates. It is an update and re-analysis of a previous study using the parsimony method. I think the questions

addressed are interesting and the manuscript is clearly written. I support its publication after a revision.

1. Include a brief description/summary of the data matrix. Mention that the characters are discrete or discretized, how many are binary and how many are multistate, and how many are ordered. I assume that the characters are variable across the taxa. Are parsimony-noninformative characters removed, and if so, how is the ascertainment bias dealt with in the program, etc.

-- More descriptions are added as suggested.

2. There should be a robustness analysis, to examine the posterior sensitivity to the priors. The model is extremely parameter-rich so that it will be hard to do a comprehensive robustness analysis, but the impact of some of the important parameters should be examined. For example, I wonder how the posterior times change when change the prior mean for the mean clock rate c by 10 folds. The $U(0, 1)$ priors on the transformed birth-death parameters look too informative and restrictive. The posterior for ν is close to 1, suggesting that the prior truncated values higher than 1.

-- We have performed additional robustness analysis for the clock rate prior as suggested, together with the previous one for root age prior. Overall, the age and rate estimates are quite robust to these prior changes. It is reasonable as the fossil ages provide good information to distinguish the times and rates in the inference thus these priors are less influential.

The reparameterization of λ and μ implies that $\lambda > \mu$, so that d ranges from 0 to infinity and v ranges from 0 to 1. In this case, a $U(0, 1)$ prior for $\nu = \mu/\lambda$ is appropriate. The same parameterization has also been used, for example, in Heath et al 2014, Gavryushkina et al 2014 and Zhang et al. 2016, for implementation convenience. MrBayes does not allow negative net diversification ($\lambda < \mu$) unfortunately.

3. The authors described the advantages of the Bayesian approach and tip dating, such as being able to integrate information from different sources. Such discussions are OK in theory, but in practice there are many caveats in such an analysis, given that the data involve a lot of uncertainties and the model has many parameters for which there is little information in the data. Basically we are asking a lot of difficult questions from very limited data (even if the dataset is the largest for the group that has ever been assembled). I would like the authors to discuss some of the caveats and take a more measured tone in the writing. For example, the use of the rock strata to form uniform bounds on the ages of the fossils involve a lot of uncertainties. The posterior estimates of times (and rates) have large intervals. This partly reflects the uncertainties in the analysis but limits the utility of the estimates.

-- We have discussed more about the uncertainties in the Bayesian tip dating approach. We note that "The stratigraphic age uncertainty represented as uniform distributions also contribute to the variances, but ignoring such uncertainty can lead to biased age estimates under the FBD process (Bardo-Sottani et al 2019)". The data do have power to reveal the heterogeneous clocks and evolutionary patterns, both in this Bayesian tip-dating and the previous parsimony analysis (Wang & Lloyd 2016). The parsimony result may look more

precise, but it only gives the most parsimonious tree (or trees with the same parsimonious length) as a point estimate. The rate estimates from ancestral state reconstructions do not account for many uncertainties either. Such estimates typically lose accuracy. Nevertheless, we state that “Further effort of coding more characters would refine the resolution”.

4. No molecular data are used in this study. Perhaps briefly comment on the advantages and disadvantages of combining molecular and morphological data in such an analysis. When you include modern species, presumably morphological characters from modern species could be used together from morphological characters based on fossils?

-- We provided a brief statement about combining molecular and morphological data. “In total-evidence dating, large amount of molecular data can typically produce reliable backbone relationships of extant taxa. But for the Mesozoic birds under study, the extant taxa would form a monophyletic group in the Paleogene thus would not contribute much to the estimation of divergence times and evolutionary rates in the stem group. Thus, abundant high-quality morphological characters and accurate stratigraphic ages of the fossils are critical.”

5. Personally I do not think partitioning is necessary: it makes the model too parameter rich to be barely usable. First, there are only a few hundred characters. Second the need to correct for the ascertainment bias means further loss of information. And thirdly, you already have a gamma model to accommodate variable rates among characters. I suggest that you look at the rate estimates for the characters and branches under the one-partition model, and average over the six regions, and confirm that the trend is the same as you see from the six-partition analysis.

-- The main aim of this study is to reveal the heterogeneous evolutionary dynamics among regions/partitions along the tree branches, which cannot be done under a single partition. We do not think this is an overparameterization issue, although at the cost of less precision. As mentioned in the results, “this gamma distribution models rate variation across characters and is independent of the gamma distributions for rate variation across branches in the relaxed clock model”. In fact, the among-character rate variation is inferred fairly homogeneous, but the among-branch rate variation is heterogeneous. Under a single partition, the branch rates are averaged across all characters, say a branch has high evolutionary rate, we don’t know which body region contributes to this high rate unless we partition the data. This cannot be done by looking at the rate for the characters alone.

6. p.7 line 134. Why is the variance for the gamma branch rate inversely proportional to the time duration of the branch (t_i). This does not make biological sense. In the geometric Brownian motion model of Thorne et al., the variance for the log rate is proportional to the time duration, so that the longer the branch the more the rate will drift.

-- We have clarified the assumption in the original model and variable transformation in the clock model section. Basically $r_i = b_i / (t_i * c)$, since b_i is gamma distributed with mean $t_i * c$ and variance $t_i * c * \sigma$, if you divide a random variable by $K = (t_i * c)$, then its mean is divided by K and its variance by K^2 . The model has variance proportional to the branch

length measured by distance, that is, the longer the branch is, the more changes (substitutions) it would accumulate.

7. The ms. will benefit from some editing.

-- We have carefully revised the writing again.